# Learn to Explore: Meta NAS via Bayesian Optimization Guided Graph Generation

## Abstract

Neural Architecture Search (NAS) automates the design of high-performing neural networks but typically targets a single predefined task, thereby restricting its real-world applicability. To address this, Meta Neural Architecture Search (Meta-NAS) has emerged as a promising paradigm that leverages prior knowledge across tasks to enable rapid adaptation to new ones. Nevertheless, existing Meta-NAS methods often struggle with poor generalization, limited search spaces, or high computational costs. In this paper, we propose a novel Meta-NAS framework, GraB-NAS. Specifically, GraB-NAS first models neural architectures as graphs, and then a hybrid search strategy is developed to find and generate new graphs that lead to promising neural architectures. The search strategy combines global architecture search via Bayesian Optimization in the search space with local exploration for novel neural networks via gradient ascent in the latent space. Such a hybrid search strategy allows GraB-NAS to discover task-aware architectures with strong performance, even beyond the predefined search space. Extensive experiments demonstrate that GraB-NAS outperforms state-of-the-art Meta-NAS baselines, achieving better generalization and search effectiveness.

## 1 Introduction

As neural networks gained popularity for their success in applications such as image classification and speech recognition, the importance of well-designed architectures became evident. To address this, Neural Architecture Search (NAS) has emerged as a paradigm for automating the design of neural networks, achieving competitive or even superior performance compared to human-crafted models across a wide range of tasks (see, e.g., Ren et al. (2021)). Despite its success, traditional NAS approaches can only enumerate in a fixed search space for a single predefined task. Nevertheless, real-world applications usually involve heterogeneous tasks, making it difficult for a single NAS model to generalize effectively. To address this challenge, recent efforts have shifted towards Meta Neural Architecture Search (Meta-NAS), where the goal is to learn an NAS model that generalizes across diverse tasks by leveraging prior knowledge to identify high-performance architectures for new, unseen tasks.

To this end, an ideal Meta-NAS framework should fulfill the following three criteria: (i) robustness - the ability to generalize across various tasks; (ii) generative capacity - the ability to synthesize novel and task-aware architectures beyond a fixed candidate set; (iii) efficiency - the capability to rapidly discover high-performing architectures for new tasks. However, most existing Meta-NAS approaches fail to fully satisfy these requirements (Shaw et al., 2019; Lee et al., 2021; Shala et al., 2023; Wang et al., 2020; Pereira et al., 2023).

To bridge the gap, we develop a novel Meta-NAS framework, termed GraB-NAS, which is robust, effective, and capable of generating novel architectures beyond the original search space. GraB-NAS employs a **hybrid search strategy** that integrates global search in the search space via Bayesian Optimization (BO) with local exploration through gradient-based optimization. Concretely, GraB-NAS models the neural architectures as graphs, and maps the given dataset and each candidate architecture (graph) in the search space to corresponding embeddings. The fused representation is then obtained and fed into a deep kernel Gaussian Process (GP) to predict the architecture's performance on the target dataset. To guide the search, BO is

employed to select promising architectures by maximizing an acquisition function over the GP surrogate. To further enhance search efficiency and explore excellent architectures, GraB-NAS performs gradient ascent on the embedding of the best-found architecture over the GP surrogate, refining it to improve the predicted performance. The updated embedding is then decoded into a new graph (architecture), which may lie outside the original search space. Therefore, this hybrid strategy enables GraB-NAS to discover high-performing and potentially novel architectures with great generalization across diverse tasks.

Extensive experiments show that GraB-NAS consistently outperforms existing Meta-NAS baselines, demonstrating better generalization across heterogeneous tasks while achieving a favorable balance between search efficiency and effectiveness.

In summary, our contributions can be concluded in three folds:

- We propose GraB-NAS, a novel Meta-NAS framework that aims to discover task-aware and high-performing architectures for unseen datasets.

- GraB-NAS is built upon a task-aware design. A hybrid search strategy is developed by combining global search via Bayesian Optimization with local exploration via gradient-based optimization, enabling architecture search beyond the predefined search space.

- Extensive experiments on multiple datasets demonstrate that the proposed GraB-NAS outperforms existing baselines, showing enhanced search effectiveness, better generalization, and stronger robustness under various scenarios.

## 2 Related Works

### 2.1 Neural Architecture Search

Neural Architecture Search (NAS) aims to automate neural network design by efficiently exploring the search space to eliminate the need for repetitive trial-and-error tuning. Existing NAS methods can be categorized into four major classes according to their search strategies: (1) reinforcement learning-based methods treat the architecture search as a sequential decision process, where a controller is trained to maximize a performance-based reward signal (e.g., accuracy) (Zoph & Le, 2016; Pham et al., 2018); (2) evolutionary algorithms iteratively evolve a population of architectures through mutation and selection based on their performance (Real et al., 2019; Lu et al., 2020b); (3) gradient-based methods relax the discrete architecture space into a continuous one, enabling efficient optimization via gradient descent (Liu et al., 2018; Cai et al., 2018; Luo et al., 2018; Dong & Yang, 2019b; Chen et al., 2020; Xu et al., 2019; Fang et al., 2020); and (4) Bayesian optimization-based approaches employ surrogate models to estimate the candidate architectures' performances and later guide the selection (White et al., 2021; Zhou et al., 2019; Kandasamy et al., 2018). While these approaches have achieved promising results, most of them are designed for predefined tasks, which limits their scalability in real-world settings involving diverse tasks. To address this, recent works have shifted towards Meta-NAS, which focuses on generalizing NAS across multiple tasks by leveraging transferable knowledge.

### 2.2 Meta Neural Architecture Search

To enable generalization across tasks, Meta Neural Architecture Search (Meta-NAS) has extended traditional NAS by leveraging prior knowledge to search for architectures suited to new tasks (Shaw et al., 2019; Lee et al., 2021; Shala et al., 2023; Wang et al., 2020; Pereira et al., 2023). Early work such as BASE formulates Meta-NAS as a Bayesian inference problem and learns a shared prior over architectures and weights to rapidly adapt to new tasks (Shaw et al., 2019). M-NAS builds a meta-learned controller to generate task-specific architectures and modulates shared meta-weights for fast adaptation to new tasks (Wang et al., 2020). Among existing works, two representative methods, MetaD2A (Lee et al., 2021) and TNAS (Shala et al., 2023), have made notable advances toward enabling transferable neural architecture search across diverse tasks.

Specifically, MetaD2A learns a task-conditioned generator that directly generates neural architectures from dataset embeddings (Lee et al., 2021). While this design enables efficient inference, it relies entirely on

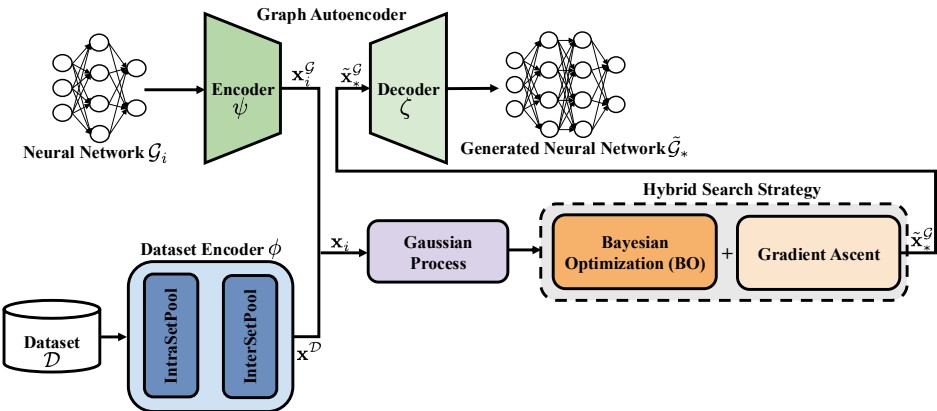

Figure 1: Overview of the GraB-NAS framework during the meta-testing stage. GraB-NAS encodes datasets and architectures, models architecture performances with a Gaussian Process, and searches for promising architectures using a hybrid search strategy combining Bayesian Optimization and gradient ascent.

a pretrained generator and predictor without leveraging evaluation feedback from the target task. As a result, it may perform poorly on out-of-distribution tasks. On the other hand, TNAS uses a deep kernel Gaussian Process to model architecture performance over dataset-architecture embeddings and applies Bayesian Optimization for architecture search (Shala et al., 2023). Although TNAS improves robustness through feedback-driven exploration, it is confined to selecting architectures from a fixed search space, which limits the discovery of novel architectures. Moreover, BO tends to converge slowly and demands substantial computational resources, reducing efficiency in practical scenarios.

To tackle these challenges, we propose GraB-NAS, a hybrid framework that effectively combines Bayesian Optimization and gradient-based optimization. GraB-NAS is robust, effective, and capable of generating novel architectures beyond the original search space.

## 3  Method

In this section, we present GraB-NAS: a **Gra**dient-assisted **B**ayesian Optimization-based Meta **N**eural **A**rchitecture **S**earch, which is designed to efficiently generate high-performing neural architectures tailored to a target dataset.

Following the standard Meta-NAS paradigm, GraB-NAS first undergoes a meta-training stage, where the framework learns transferable knowledge from a collection of training tasks. During the meta-testing stage, the prior knowledge learned is exploited to rapidly adapt the search process to an unseen dataset, enabling efficient discovery of high-performing architectures.

When adapting to a new dataset, GraB-NAS employs a hybrid search strategy that combines global search in the search space via Bayesian Optimization (BO) with local exploration through gradient-based optimization. Specifically, BO operates on a Gaussian Process (GP) surrogate model that predicts architecture performance from cross-modal embeddings of dataset–architecture pairs. The surrogate model later guides the selection of promising candidates by maximizing an acquisition function. To further improve efficiency and effectiveness, we introduce a gradient ascent step: for the top-performing candidate found hitherto, we perform gradient ascent in the architecture latent space to increase its GP-predicted performance. The refined embedding is then decoded into a potentially novel architecture, enabling exploration beyond the fix search space. In this way, the joint optimization scheme strikes a balance between global search and local exploration, facilitating the discovery of high-quality architectures. An overview of the GraB-NAS framework is provided in Figure 1. The subsequent subsections detail the search procedure and core components of our method.

### 3.1 Gradient-assisted Bayesian Optimization

GraB-NAS aims to rapidly discover high-performing neural networks given a dataset $\mathcal{D}$. To incorporate dataset-specific information into the search process, we first encode $\mathcal{D}$ using a dataset encoder $\phi(\mathcal{D})$, which produces a latent embedding $\mathbf{x}^{\mathcal{D}}$ that captures its structural and semantic characteristics (Section 3.2). Each candidate architecture, typically drawn from a predefined search space (e.g., NAS-Bench-201), is modeled as a directed acyclic graph (DAG) $\mathcal{G}_i = (\mathcal{V}_i, \mathcal{E}_i)$, where $\mathcal{V}_i$ denotes the node set, with each node $v \in \mathcal{V}_i$ representing computational operations, and each edge $e \in \mathcal{E}_i$ represents directed data flow between nodes. A graph encoder is then employed to map $\mathcal{G}_i$ to a latent embedding $\mathbf{x}_i^{\mathcal{G}}$ by incorporating both topological and operational information (Section 3.3.1). A deep kernel Gaussian Process (GP) is adopted as a surrogate model to estimate the performance of a neural network $\mathcal{G}_i$ on a dataset $\mathcal{D}$. To this end, a multilayer perceptron (MLP) is used to obtain a fused representation $\mathbf{x}_i$ based on the concatenated dataset and architecture embeddings:

$$\mathbf{x}_i = \text{MLP}(\mathbf{x}^{\mathcal{D}}, \mathbf{x}_i^{\mathcal{G}}), \tag{1}$$

which serves as the input to the GP surrogate (Wilson et al., 2016). Then, GP is employed to learn the black-box function $\pi(\cdot)$, which maps the fused embedding $\mathbf{x}_i$ to the corresponding architecture performance $\pi(\mathbf{x}_i)$.

Specifically, given a set of fused representations $\{\mathbf{x}_1, \ldots, \mathbf{x}_n\}$ and their outputs $\{\pi(\mathbf{x}_1), \ldots, \pi(\mathbf{x}_n)\}$. For a new fused representation $\mathbf{x}'$, GP assumes that the outputs jointly follow a multivariate Gaussian distribution (Schulz et al., 2018):

$$\begin{bmatrix} \pi(\mathbf{x}_1) \\ \vdots \\ \pi(\mathbf{x}_n) \\ \pi(\mathbf{x}') \end{bmatrix} \sim \mathcal{N}\left( \mathbf{0}, \begin{bmatrix} \kappa_{1,1} & \cdots & \kappa_{1,n} & \kappa_{1,\prime} \\ \vdots & \ddots & \vdots & \vdots \\ \kappa_{n,1} & \cdots & \kappa_{n,n} & \kappa_{n,\prime} \\ \kappa_{\prime,1} & \cdots & \kappa_{\prime,n} & \kappa_{\prime,\prime} \end{bmatrix} \right), \tag{2}$$

where $\kappa_{r,s} := k(\mathbf{x}_r, \mathbf{x}_s)$ and $\kappa_{r,\prime} := k(\mathbf{x}_r, \mathbf{x}')$, with $k(\cdot, \cdot)$ being the kernel function (e.g., Matérn) that measures the similarity between the data points (Williams & Rasmussen, 2006). Conditioning on the observed outputs yields a Gaussian posterior over $\pi(\mathbf{x}')$, with predictive mean and variance (Schulz et al., 2018):

$$\mu(\hat{\pi}(\mathbf{x}')) = \mathbf{k}'^{\top} \mathbf{K}^{-1} \begin{bmatrix} \pi(\mathbf{x}_1) \\ \vdots \\ \pi(\mathbf{x}_n) \end{bmatrix}, \tag{3}$$

$$\sigma^2(\hat{\pi}(\mathbf{x}')) = k(\mathbf{x}', \mathbf{x}') - \mathbf{k}'^{\top} \mathbf{K}^{-1} \mathbf{k}', \tag{4}$$

where $\mathbf{K} \in \mathbb{R}^{n \times n}$ is the kernel matrix with entries $\mathbf{K}_{rs} = k(\mathbf{x}_r, \mathbf{x}_s)$, and $\mathbf{k}' = [k(\mathbf{x}_1, \mathbf{x}'), \ldots, k(\mathbf{x}_n, \mathbf{x}')]^{\top}$.

To explore the search space, we develop a **hybrid search strategy** that combines global search via Bayesian Optimization with local exploration via gradient-based optimization in the latent space. Initially, the GP surrogate is trained according to Eq. 3 and Eq. 4 on a small support set $\mathcal{S}$ consisting of support data pairs (i.e., $\{\mathcal{G}_i, \pi(\mathbf{x}_i)\}$), where each $\mathcal{G}_i$ is one of the top-$B$ ($B$ is a hyperparameter) best-performing architectures on the training dataset, and the corresponding $\pi(\mathbf{x}_i)$ is its performance evaluated on the target dataset.

#### 3.1.1 Bayesian Optimization-based Search

At each search iteration, BO selects the most promising candidate by maximizing an acquisition function (e.g., Expected Improvement) based on the GP surrogate (Snoek et al., 2012):

$$\text{EI}(\hat{\pi}(\mathbf{x}_i)) = \mathbb{E}[\max(0, \hat{\pi}(\mathbf{x}_i) - \pi(\mathbf{x}_*))], \tag{5}$$

where $\pi(\mathbf{x}_*)$ is the best observed performance in the current support set. The selected architecture is then evaluated on the target dataset $\mathcal{D}$, and the corresponding architecture-performance pair is added to $\mathcal{S}$ to further refine the GP surrogate, thereby improving the predictive accuracy.

However, relying solely on BO introduces two pivotal limitations. First, BO typically requires multiple evaluations before the GP surrogate becomes sufficiently accurate to guide the search effectively, resulting

in slow convergence. Second, the BO-based search is inherently restricted to the predefined search space, limiting its ability to generate novel architectures.

### 3.1.2 Gradient-based Exploration

To improve search efficiency and effectiveness, and address the aforementioned limitations of BO, we introduce a gradient ascent exploration scheme. Concretely, among the previously evaluated architectures in the support set $\mathcal{S}$, we identify the architecture $\mathcal{G}_*$ with the highest observed performance $\pi(\mathbf{x}_*)$. Gradient ascent is then applied to its latent embedding $\mathbf{x}_*^{\mathcal{G}}$ to increase the GP-predicted performance:

$$\tilde{\mathbf{x}}_*^{\mathcal{G}} = \mathbf{x}_*^{\mathcal{G}} + \eta \nabla_{\mathbf{x}_*^{\mathcal{G}}} \mu(\hat{\pi}(\mathbf{x}_*)), \tag{6}$$

where $\eta$ denotes the learning rate and $\mu(\hat{\pi}(\mathbf{x}_*))$ denotes the GP's posterior mean prediction (see Eq. 3). The updated embedding $\tilde{\mathbf{x}}_*^{\mathcal{G}}$ is subsequently decoded into a new architecture $\tilde{\mathcal{G}}_*$ using a graph decoder (Section 3.3.2). This new architecture is expected to have improved performance. The new architecture-performance pair is then added to the support set $\mathcal{S}$. Importantly, the gradient-based exploration also introduces the potential for generating novel architectures.

### 3.1.3 Hybrid Search Strategy

Combining the BO-based and gradient-based strategies, GraB-NAS adopts a hybrid search strategy that alternates between BO-based global search and gradient-based local exploration. Specifically, BO is employed for $T_{\mathrm{BO}}$ rounds to warm up the GP surrogate. Thereafter, both the BO-based search and gradient-based exploration are employed during each search iteration. Overall, such a hybrid search strategy leverages the complementary strengths of Bayesian Optimization and gradient ascent. This synergy strikes a better balance between search efficiency and effectiveness, and promotes the discovery of both innovative and performant architectures. The following subsections will provide further information regarding the dataset encoder and graph autoencoder employed in the proposed framework.

### 3.2 Dataset Encoder

To facilitate task-specific architecture search, we first encode the target dataset $\mathcal{D}$ into an embedding $\mathbf{x}^{\mathcal{D}}$ that captures its underlying structures and semantics. This embedding is designed to reflect both the distribution of instances within each class and the inter-class relationships. In doing so, it provides task-specific context for the architecture search, allowing the proposed framework to generalize effectively across diverse tasks.

We adopt a dataset encoder $\phi(\mathcal{D})$ to obtain the dataset embedding:

$$\mathbf{x}^{\mathcal{D}} = \phi(\mathcal{D}). \tag{7}$$

The encoder is implemented as two hierarchically stacked *Set Transformer* modules (Lee et al., 2019; 2021) (see Appendix A for details). The first module processes randomly sampled instances from each class to produce class-level prototypes, while the second module aggregates these prototypes into a dataset-level embedding. Both modules employ permutation-invariant, attention-based pooling layers, ensuring robustness to instance order and enabling the encoder to capture both intra-class features and higher-level inter-class relations. This hierarchical embedding provides a compact yet informative context for guiding architecture search on the target dataset.

### 3.3 Graph Autoencoder

GraB-NAS employs a graph autoencoder, which consists of an encoder maps discrete neural networks (modeled as DAGs) into a continuous latent space and a decoder reconstructs them back.

Specifically, the encoder $\psi(\mathcal{G})$ transforms each candidate architecture $\mathcal{G}_i$ into a latent vector $\mathbf{x}_i^{\mathcal{G}}$ by processing its nodes in topological order and incorporating both structural and operational properties. During meta-testing, the decoder $\zeta(\mathbf{x}^{\mathcal{G}})$ then takes a refined embedding $\tilde{\mathbf{x}}_*^{\mathcal{G}}$ obtained via gradient ascent as input and reconstructs a corresponding architecture $\tilde{\mathcal{G}}_*$ from it.

### 3.3.1 Graph Encoder

To embed a neural network $\mathcal{G}_i$ in a continuous latent space, we adopt the graph encoder $\psi(\mathcal{G})$ proposed in D-VAE (Variational Autoencoder for Directed Acyclic Graphs) (Zhang et al., 2019):

$$\mathbf{x}_i^{\mathcal{G}} = \psi(\mathcal{G}_i). \tag{8}$$

The encoder encodes node operations and their topological dependencies using a gated recurrent unit (GRU)-based message-passing mechanism, followed by bidirectional aggregation to capture both forward and backward structural information. The resulting embedding $\mathbf{x}_i^{\mathcal{G}}$ represents the architecture's topology and operation types, enabling accurate performance prediction via the GP surrogate and gradient-based optimization in the latent space (see Appendix B for details).

### 3.3.2 Graph Decoder

The graph decoder $\zeta(\mathbf{x}^{\mathcal{G}})$ is employed to reconstruct neural networks $\tilde{\mathcal{G}}_*$ from refined latent embeddings $\tilde{\mathbf{x}}_*^{\mathcal{G}}$, in order to generate architectures that may yield better performance (Zhang et al., 2019):

$$\tilde{\mathcal{G}}_* = \zeta(\tilde{\mathbf{x}}_*^{\mathcal{G}}). \tag{9}$$

The decoder generates a graph iteratively. Starting from the embedding $\tilde{\mathbf{x}}_*^{\mathcal{G}}$, the decoder adds one node at a time. At each step, the decoder predicts the operation type of the new node and determines its connections to previously generated nodes. Details of the graph decoder can be found in Appendix B. By decoding refined embeddings $\tilde{\mathbf{x}}_*^{\mathcal{G}}$, $\zeta(\mathbf{x}^{\mathcal{G}})$ enables GraB-NAS to explore architectures beyond the predefined search space, enhancing both search efficiency and the likelihood of discovering novel, high-performing network designs.

### 3.4 Meta-training

During meta-training, the goal is to learn generalizable dataset and graph encoders, along with the MLP. This is accomplished by maximizing the log marginal likelihood of the GP surrogate on the meta-training set:

$$\arg\max_{w} \log p(\boldsymbol{\pi} \mid \mathbf{X}(w)) \propto$$
$$\arg\min_{w} \left[ \boldsymbol{\pi}^{\top} \mathbf{K}^{-1}(\mathbf{X}(w))\boldsymbol{\pi} + \log\det(\mathbf{K}(\mathbf{X}(w))) \right], \tag{10}$$

where $w$ denotes the trainable weights, $\mathbf{X} = [\mathbf{x}_1, \ldots, \mathbf{x}_m]^{\top}$ denotes the joint representations of dataset-architecture pairs in the meta-training dataset, and $\boldsymbol{\pi} = [\pi(\mathbf{x}_1), \ldots, \pi(\mathbf{x}_m)]^{\top}$ represents the corresponding observed performances.

The graph decoder is trained separately after the encoders are trained. Specifically, it is trained via the teacher forcing scheme in (Zhang et al., 2019) using a small subset of architecture embeddings generated by the trained graph encoder, paired with their corresponding architectures.

### 3.5 Meta-testing

During meta-testing, GraB-NAS exploits the knowledge acquired during meta-training to adapt the search to a new target dataset. After encoding the dataset and candidate architectures using the trained encoders, the search proceeds in two coordinated steps: BO selects candidate architectures from the search space by maximizing the acquisition function over the GP surrogate; and gradient ascent refines the best architecture's embedding to explore beyond the fixed search space for better suited neural networks. The complete procedure is summarized in Algorithm 1.

## 4 Experiment

To validate the effectiveness and efficiency of GraB-NAS, we conducted extensive experiments on multiple benchmark datasets and compared the proposed method against baselines. Moreover, we validated our design choice and performed ablation studies. The goal is to assess GraB-NAS's search effectiveness and efficiency, robustness, and generative capability.

---

**Algorithm 1** Meta-Test Procedure of GraB-NAS

---

**Input**: Target dataset $\mathcal{D}$, candidate architectures $\{\mathcal{G}_i\}$ in the predefined search space, pretrained dataset encoder $\phi(\mathcal{D})$, graph encoder $\psi(\mathcal{G})$, graph decoder $\zeta(\mathbf{x}^{\mathcal{G}})$
**Parameter**: number of search iterations $T$, number of initial support data pairs $B$, number of BO-only search iterations $T_{\text{BO}}$, learning rate $\eta$
**Output**: Best architecture $\mathcal{G}_{\text{best}}$ for the target dataset $\mathcal{D}$
**Notation**: $\pi(\mathbf{x}_i)$ denotes the actual performance of architecture $\mathcal{G}_i$ on $\mathcal{D}$

 1: Encode dataset: $\mathbf{x}^{\mathcal{D}} = \phi(\mathcal{D})$
 2: Encode: $\mathbf{x}_i^{\mathcal{G}} = \psi(\mathcal{G}_i), \quad \forall i$
 3: Fuse: $\mathbf{x}_i = \text{MLP}(\mathbf{x}^{\mathcal{D}}, \mathbf{x}_i^{\mathcal{G}}), \quad \forall i$
 4: Initialize support set $\mathcal{S} = \{(\mathcal{G}_i, \pi(\mathbf{x}_i))\}$ with the architectures from $\{\mathcal{G}_i\}$ that achieved the top $B$ average performance across meta-training tasks, where $\pi(\mathbf{x}_i)$ denotes the actual performance evaluated on $\mathcal{D}$
 5: **for** $t = 1$ to $T$ **do**
 6: $\quad$ Update $\mu$ and $\sigma^2$ in GP with $\mathcal{S}$ via Eq. 3 and Eq. 4
 7: $\quad$ Select $\mathcal{G}_t \in \{\mathcal{G}_i \notin \mathcal{S}\}$ by maximizing the acquisition function via Eq. 5
 8: $\quad$ Evaluate actual performance $\pi(\mathbf{x}_t)$ of $\mathcal{G}_t$ on $\mathcal{D}$
 9: $\quad$ Update support set: $\mathcal{S} \leftarrow \mathcal{S} \cup \{(\mathcal{G}_t, \pi(\mathbf{x}_t))\}$
10: $\quad$ **if** $t > T_{\text{BO}}$ **then**
11: $\quad\quad$ Identify $\mathcal{G}_* = \arg\max_{\mathcal{G}_j \in \mathcal{S}} \pi(\mathbf{x}_j)$
12: $\quad\quad$ Update GP with $\mathcal{S}$ via Eq. 3 and Eq. 4
13: $\quad\quad$ Gradient ascent: $\tilde{\mathbf{x}}_*^{\mathcal{G}} = \mathbf{x}_*^{\mathcal{G}} + \eta \nabla_{\mathbf{x}_*^{\mathcal{G}}} \mu(\hat{\pi}(\mathbf{x}_*))$
14: $\quad\quad$ Decode: $\tilde{\mathcal{G}}_* = \zeta(\tilde{\mathbf{x}}_*^{\mathcal{G}})$
15: $\quad\quad$ **if** $\tilde{\mathcal{G}}_* \notin \mathcal{S}$ **then**
16: $\quad\quad\quad$ Evaluate the performance of $\tilde{\mathcal{G}}_*$ on $\mathcal{D}$
17: $\quad\quad\quad$ Update support set $\mathcal{S} \leftarrow \mathcal{S} \cup \{(\tilde{\mathcal{G}}_*, \pi(\tilde{\mathbf{x}}_*))\}$, where $\tilde{\mathbf{x}}_* = \text{MLP}(\mathbf{x}^{\mathcal{D}}, \tilde{\mathbf{x}}_*^{\mathcal{G}})$
18: $\quad\quad$ **end if**
19: $\quad$ **end if**
20: **end for**
21: **return** $\mathcal{G}_{\text{best}} = \arg\max_{\mathcal{G}_j \in \mathcal{S}} \pi(\mathbf{x}_j)$

---

## 4.1 Datasets and Experimental Setup

**NAS-Bench-201 Search Space.** The NAS-Bench-201 is a benchmark designed for evaluating NAS algorithms (Dong & Yang, 2020). It consists of architectures sharing the same macro structure, which is a stack of repeated cells connected in series. Each cell is represented as a DAG with four nodes, where each directed edge denotes an operation selected from a fixed set of five candidates: zeroize, skip connection, 1-by-1 convolution, 3-by-3 convolution, and 3-by-3 average pooling.

**Datasets.** Following the MetaD2A setting (Lee et al., 2021), we construct our training dataset using subsets of ImageNet-1K (Deng et al., 2009). For evaluation, we apply the proposed GraB-NAS to six unseen datasets: CIFAR-10 (Krizhevsky et al., 2009), CIFAR-100 (Krizhevsky et al., 2009), MNIST (LeCun, 1998), SVHN (Netzer et al., 2011), Aircraft (Maji et al., 2013), and Oxford-IIIT Pet (Parkhi et al., 2012). The candidate architectures are trained from scratch using the NAS-Bench-201 pipeline for consistency.

**Baselines.** We compare GraB-NAS with three categories of baselines: Random Search, state-of-the-art (SOTA) NAS methods, and Meta-NAS methods. We apply Random Search by uniformly sampling architectures from the search space. For SOTA NAS, we include several strong one-shot methods that have demonstrated competitive performance, namely GDAS (Dong & Yang, 2019b), SETN (Dong & Yang, 2019a), PC-DARTS (Xu et al., 2019), and DrNAS (Chen et al., 2020). For these methods, we report the published results on the benchmark datasets. For the Meta-NAS methods, we consider the two most relevant approaches, MetaD2A (Lee et al., 2021) and TNAS (Shala et al., 2023). MetaD2A uses a meta-trained decoder to generate neural networks conditioned on the target dataset, and employs a performance predictor to rank the generated candidates for final selection (Lee et al., 2021). TNAS employs a meta-learned Bayesian

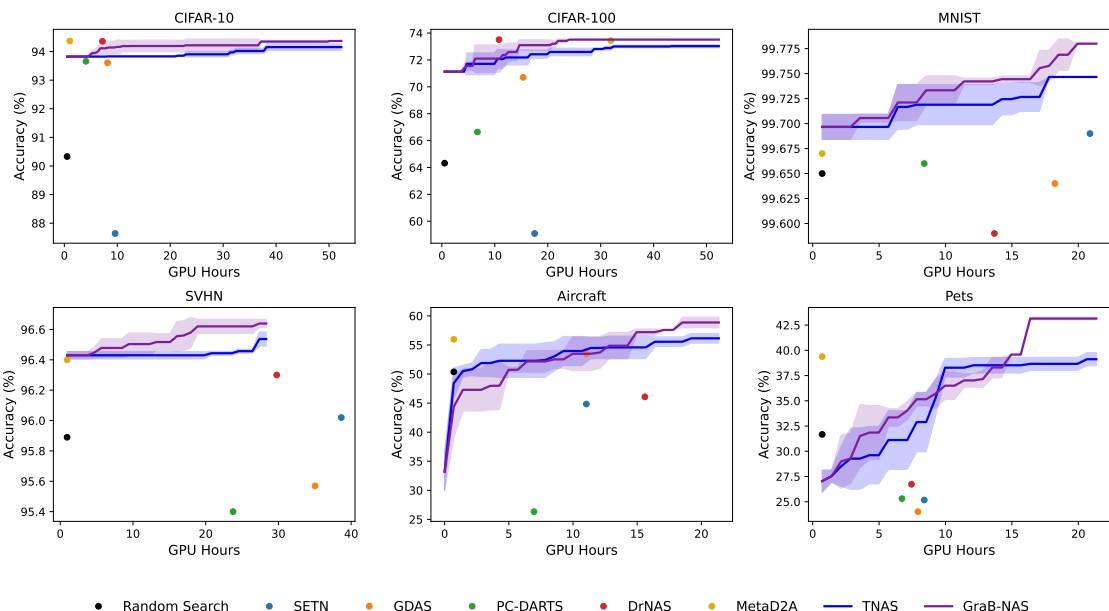

Figure 2: Performance on unseen datasets in the meta-testing stage.

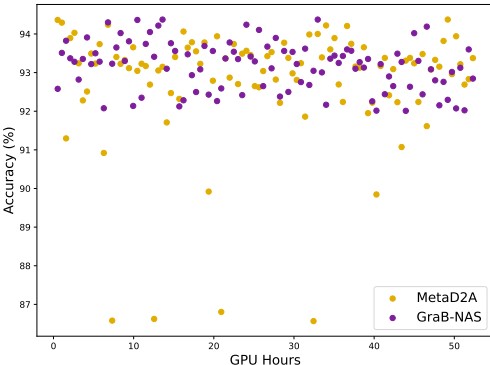

Figure 3: Comparison of MetaD2A and GraB-NAS in terms of accuracy of the generated/selected architectures on CIFAR-10. During the meta-test, MetaD2A evaluates the generated architectures based on the ranks produced by its predictor, whereas GraB-NAS employs a meta-learned surrogate to decide which architectures to evaluate sequentially.

surrogate to search for high-performing architectures for the target task (Shala et al., 2023). All reported results are averaged over three runs. We will release our code to facilitate reproducibility.

## 4.2 Results on Unseen Datasets

We evaluated the proposed GraB-NAS across six diverse image classification datasets, comparing it against Random Search, SOTA NAS methods, and Meta-NAS approaches. Detailed hyperparameter settings are summarized in Appendix C. As shown in Figure 2, we present the test accuracy versus computational cost for all methods and datasets; detailed numerical results are provided in Appendix D. Overall, GraB-NAS consistently outperforms all baselines in final accuracy. We also provide evaluation results beyond the NAS-Bench-201 search space in Appendix F.

Compared to Random Search, GraB-NAS excels in search performance under most circumstances. Although Random Search may show competitive results during the initial stages on Aircraft and Pets, it lacks theoretical

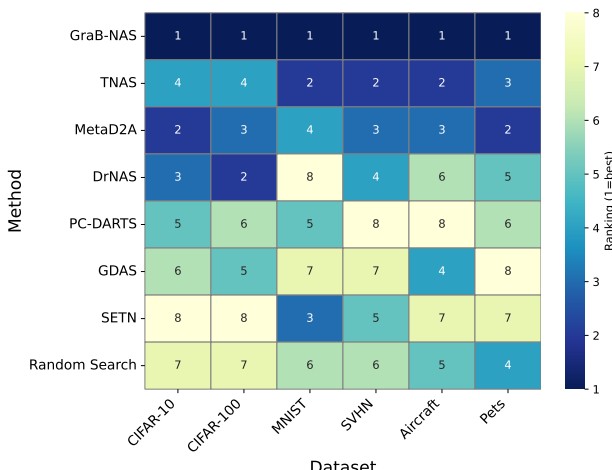

Figure 4: Performance ranking across 6 datasets.

grounding and relies on unguided exploration in the search space. As evidenced in Appendix D, Random Search's performances exhibit large standard deviations, indicating poor consistency and limited robustness across different runs. On the other hand, the results demonstrate that GraB-NAS achieves improvements over existing NAS methods in both anytime performance and final performance across the majority of benchmarks, with particularly significant gains in fine-grained datasets (Aircraft and Pets).

In comparison to MetaD2A, while MetaD2A is capable of identifying a reasonably good neural network, it consistently underperforms GraB-NAS in terms of final accuracy. This may arise from the fact that MetaD2A passively follows the knowledge acquired during meta-training without incorporating any feedback from the target dataset during meta-testing. In contrast, GraB-NAS leverages a task-aware surrogate to iteratively adapt to the target dataset, enabling the discovery of higher-quality architectures. As illustrated in Figure 3, the neural architectures generated by GraB-NAS generally outperform those obtained by MetaD2A. On average, GraB-NAS achieves higher and more stable accuracy results compared to MetaD2A, and MetaD2A also exhibits occasional outliers. This becomes particularly pronounced when evaluating on datasets that are poorly aligned with the meta-training datasets, where the absence of task-specific knowledge hinders the searching. Besides, Figure 2 demonstrates that the proposed GraB-NAS achieves overall stronger anytime and final performance on most datasets than TNAS, achieving better search effectiveness and a more favorable balance between search efficiency and effectiveness. Further performance comparisons between GraB-NAS and TNAS/MetaD2A are provided in Appendix H.

To further evaluate cross-dataset performance, we visualize the rankings of each method across the six datasets in Figure 4. As expected, Meta-NAS approaches generally achieve better average rankings compared to conventional NAS methods, highlighting their ability to leverage prior knowledge for improved generalization. Notably, while completing methods show fluctuations in rankings across datasets, GraB-NAS consistently achieves the best performance across all datasets, indicating stronger consistency and better generalization capability.

In summary, GraB-NAS demonstrates the ability to identify high-quality architectures across various datasets, sometimes achieving great results even under limited search budgets. By combining Bayesian Optimization with gradient-based optimization, GraB-NAS enhances search effectiveness while promoting better generalization across heterogeneous tasks. These advantages position GraB-NAS as a practical and reliable solution for neural architecture search in real-world settings.

## 4.3 Comparison of GP Surrogates

To further investigate the effectiveness and efficiency of our design choice, we compared the exact GP employed in GraB-NAS with two scalable alternatives: Sparse Gaussian Process (Sparse GP) and Random

Table 1: Computational complexity of different GP surrogates.

| Surrogate | Computational Complexity |
| --- | --- |
| Exact GP | $O(|\mathcal{S}|^3)$ |
| Sparse GP | $O(|\mathcal{S}|^2 M)$ |
| RFGP | $O(|\mathcal{S}|D^2)$ |

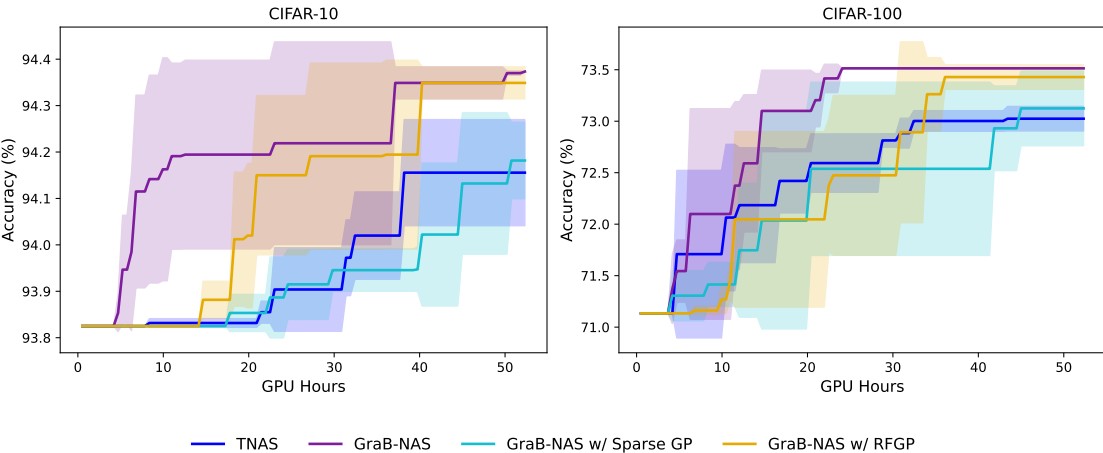

Figure 5: Performance of different surrogates on unseen datasets in the meta-testing stage.

Feature Gaussian Process (RFGP) (Leibfried et al., 2020; Lu et al., 2020a). Although the exact GP achieves accurate modeling of the architecture-performance relationship, its cubic complexity $O(|\mathcal{S}|^3)$ may incur higher computational costs and hinder scalability as the number of support data pairs increases. This section aims to evaluate whether the approximate GP variants can maintain comparable performance while reducing runtime overhead. Specifically, Sparse GP introduces a small set of $M$ inducing points to form a low-rank approximation of the kernel matrix (Leibfried et al., 2020), whereas RFGP replaces the kernel function with $D$ random features (Lu et al., 2020a). Table 1 summarizes the computational cost of the three surrogates, and the detailed hyperparameter settings are provided in Appendix C. Besides, computational cost of each component is provided in Appendix E.

Figure 5 presents the test accuracy over GPU hours on CIFAR-10/100 for three surrogate choices. Although both Sparse GP and RFGP substantially reduce the per search iteration computational cost, GraB-NAS with these surrogates exhibits noticeably weaker search performance. Across both datasets, GraB-NAS with the exact GP consistently achieves higher accuracy under the same computational budget, demonstrating superior searching performance. Overall, these results validate our design choice of using the exact GP as the surrogate in GraB-NAS as it offers a superior trade-off between modeling fidelity and computational efficiency, making the proposed method an effective and efficient Meta-NAS framework.

## 4.4  Ablation Studies

To validate the effectiveness of our method, we conducted an ablation study on the gradient ascent step and the graph decoder. Specifically, we first evaluate a GraB-NAS w/o gradient ascent, which removes gradient-based optimization in the latent space and relies solely on BO to select candidate architectures. While this BO-only variant is able to identify competitive architectures, it consistently underperforms the full GraB-NAS on both datasets. Note that the differences between the TNAS and the GraB-NAS w/o gradient ascent come from some design choices. We further compare the proposed GraB-NAS with a variant denoted as GraB-NAS w/ KNN, where the graph decoder is replaced by a *k*-nearest neighbors (KNN) mapping the refined latent embeddings to neural networks. As shown in Table 2, this variant already outperforms TNAS and GraB-NAS w/o gradient ascent on both CIFAR-10 and CIFAR-100, demonstrating that the

Table 2: Ablation study on CIFAR-10 and CIFAR-100. *Note*: All units are in %.

| Method | CIFAR-10 | CIFAR-100 |
|---|---|---|
| MetaD2A | **94.37** $\pm$ 0.00 | 73.44 $\pm$ 0.05 |
| TNAS | 94.16 $\pm$ 0.11 | 73.02 $\pm$ 0.12 |
| GraB-NAS w/o gradient ascent | **94.37** $\pm$ 0.00 | 73.43 $\pm$ 0.12 |
| GraB-NAS w/ KNN | 94.28 $\pm$ 0.13 | 73.43 $\pm$ 0.12 |
| GraB-NAS w/ decoder | **94.37** $\pm$ 0.00 | **73.51** $\pm$ 0.00 |

gradient-based optimization in the latent space contributes to improved search effectiveness. When the decoder is introduced (GraB-NAS w/ decoder), we observe further performance gains and GraB-NAS achieves SOTA results, which indicates that the decoder enables the proposed framework to explore beyond the constrained search space.

Table 3: Performance with pruned search spaces on CIFAR-10/100 datasets. *Note*: All units are in %. "Remaining Best" indicates the highest achievable accuracy in the pruned search space. $\Delta$ denotes the gap between the accuracy of the best architecture found by TNAS/GraB-NAS and the remaining best architecture in the pruned space.

| Method | Pruning | Remaining Best | TNAS Acc ($\Delta \uparrow$) | GraB-NAS Acc ($\Delta \uparrow$) |
|---|---|---|---|---|
| CIFAR-10 | Top-10 removed | 94.22 | 94.09 $\pm$ 0.09 ($-0.13$) | **94.37** $\pm$ 0.00 ($+0.15$) |
| | Top-20 removed | 94.12 | 94.02 $\pm$ 0.01 ($-0.10$) | **94.28** $\pm$ 0.13 ($+0.16$) |
| | Top-50 removed | 93.95 | 93.89 $\pm$ 0.00 ($-0.06$) | **94.22** $\pm$ 0.20 ($+0.27$) |
| CIFAR-100 | Top-10 removed | 72.88 | 72.82 $\pm$ 0.07 ($-0.06$) | **73.43** $\pm$ 0.12 ($+0.55$) |
| | Top-20 removed | 72.65 | 72.39 $\pm$ 0.15 ($-0.26$) | **73.20** $\pm$ 0.45 ($+0.55$) |
| | Top-50 removed | 72.03 | 71.83 $\pm$ 0.24 ($-0.20$) | **73.11** $\pm$ 0.57 ($+1.08$) |

### 4.5 Effectiveness of GraB-NAS

We further evaluate GraB-NAS's effectiveness under constrained conditions by pruning the search space. Specifically, we progressively remove the top-performing architectures from the original search space to create inferior search spaces and assess GraB-NAS and TNAS's abilities to discover high-quality architectures. As highlighted in Table 3, GraB-NAS consistently discover architectures that surpass the best remaining candidates, despite the reduced quality of the search space. This again demonstrates the strength of the decoder, which enables GraB-NAS to explore beyond the limited search space. In contrast, TNAS is confined to searching only within the pruned search space and can therefore, at best, retrieve architectures that are inferior to the globally best-performing ones.

## 5 Conclusion

In this paper, we propose GraB-NAS, a novel Meta Neural Architecture Search framework that synergizes global search via Bayesian Optimization with local exploration through gradient-based optimization to achieve effective architecture search. Using a dataset encoder, a graph autoencoder, and a deep kernel Gaussian Process surrogate, GraB-NAS is capable of discovering high-performance and task-aware architectures, which are even *beyond* the predefined search space. Through extensive experiments on diverse and unseen datasets, we demonstrated that GraB-NAS consistently outperforms existing NAS baselines in final accuracy. Notably, the GraB-NAS framework exhibits strong generalization, robust search effectiveness, and adaptability under constrained search spaces. These results suggest that GraB-NAS provides a practical and scalable solution for NAS in real-world, cross-task scenarios.

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

## A    Set Transformer Modules

Following Lee et al. (2021), we adopt the *Set Transformer* as the core building block of the dataset encoder (Lee et al., 2019), leveraging its ability to process unordered sets while modeling higher-order interactions among elements via attention.

We employ two key components: the Set Attention Block (SAB) and Pooling by Multi-head Attention (PMA). The SAB learns contextualized features for each element in the input set using self-attention, while the PMA aggregates a set of input features into a fixed number of output vectors using learnable seed vectors.

Given an input set $\mathbf{Y} \in \mathbb{R}^{\alpha \times \beta}$, SAB applies multi-head self-attention followed by a residual connection and a feedforward network:

$$\text{SAB}(\mathbf{Y}) = \text{LN}(\mathbf{H} + \text{MLP}(\mathbf{H})),$$
$$\text{where} \quad \mathbf{H} = \text{LN}(\mathbf{Y} + \text{MH}(\mathbf{Y}, \mathbf{Y}, \mathbf{Y})). \tag{11}$$

Here, $\text{MH}(\cdot)$ denotes multi-head attention, and $\text{LN}(\cdot)$ is layer normalization. Furthermore, to obtain a fixed-size output embedding, we apply PMA on the set $\mathbf{Y}$ encoded by SAB using $\gamma$ learnable seed vectors $\mathbf{R} \in \mathbb{R}^{\gamma \times \beta}$:

$$\text{PMA}(\mathbf{Y}) = \text{LN}(\mathbf{H} + \text{MLP}(\mathbf{H})),$$
$$\text{where} \quad \mathbf{H} = \text{LN}(\mathbf{R} + \text{MH}(\mathbf{R}, \text{MLP}(\mathbf{Y}), \text{MLP}(\mathbf{Y}))). \tag{12}$$

In our setting, we set $\gamma = 1$ to produce a single vector embedding.

## B    Graph Autoencoder

In this paper, we implement a graph autoencoder to map discrete neural networks to the latent space and reconstruct new architectures from it.

**Graph Encoder.**    We adopt the graph encoder proposed in D-VAE (Zhang et al., 2019). For a neural network $\mathcal{G}_i$, let a one-hot vector $\mathbf{o}_v$ denote node $v$'s operation type and $\mathbf{h}_v$ be its hidden state. The hidden state $\mathbf{h}_v$ is computed using a gated recurrent unit (GRU) (Cho et al., 2014), which takes the operation type $\mathbf{o}_v$ and the aggregated message from node $v$'s predecessors as input:

$$\mathbf{h}_v = \text{GRU}_{\text{enc}}(\mathbf{o}_v, \mathbf{h}_v^{\text{in}}). \tag{13}$$

Here, the incoming message $\mathbf{h}_v^{\text{in}}$ is computed as:

$$\mathbf{h}_v^{\text{in}} = \sum_{u \to v} g(\mathbf{h}_u) \odot m(\mathbf{h}_u), \tag{14}$$

where $\odot$ denotes element-wise multiplication, and $m(\cdot)$ and $g(\cdot)$ are learnable mapping and gating networks, respectively. The hidden state of the last node in the topological order is used as the embedding for the DAG. To enrich the architecture embedding, we apply reverse message passing using another GRU in the opposite direction. The final hidden states from both directions are then concatenated and passed through an MLP to obtain the architecture embedding $\mathbf{x}_i^{\mathcal{G}}$.

**Graph Decoder.** Decoding begins by projecting $\tilde{\mathbf{x}}_*^{\mathcal{G}}$ through an MLP followed by *tanh* to obtain the initial hidden state $h_0$. The decoder $\zeta(\mathbf{x}^{\mathcal{G}})$ then generates the neural network sequentially, producing one node at a time. At each step, a new node $v_k$ is added by predicting its operation type based on the current graph state $\mathbf{h}_g := \mathbf{h}_{v_{k-1}}$:

$$\mathbf{o}_{v_k} = f_{\text{node}}(\mathbf{h}_g), \tag{15}$$

where $f_{\text{node}}$ is an MLP followed by *softmax*. If $\mathbf{o}_{v_k}$ corresponds to the end-of-graph, decoding terminates and all leaf nodes are connected to $v_k$.

Otherwise, the hidden state of $v_k$ is updated using a GRU:

$$\mathbf{h}_{v_k} = \text{GRU}_{\text{dec}}(\mathbf{o}_{v_k}, \mathbf{h}_{v_k}^{\text{in}}), \tag{16}$$

where $\mathbf{h}_{v_k}^{\text{in}}$ is computed using the same message aggregation scheme as in Eq. 14.

To determine the connectivity between previously generated nodes $\{v_l\}_{l=1}^{k-1}$ and node $v_k$, the edge probability is computed as:

$$e_{l \to k} = f_{\text{edge}}(\mathbf{h}_{v_l}, \mathbf{h}_{v_k}), \tag{17}$$

where $f_{\text{edge}}$ is an MLP followed by *sigmoid*. The edges are sampled sequentially in the order $\{v_{k-1}, \ldots, v_1\}$, and each newly added edge triggers a hidden state update for $v_k$ based on Eq. 16.

## C Experimental Details

We chose $B = 5$, $T_{BO} = 10$, and $\eta = 1$ in our experiments. Specifically, $M$ in the Sparse GP and $D$ in the RFGP were adjusted according to $T$: for $T$ fewer than 10, the models used 2 inducing points or features; for $T$ between 10 and 20, 4; between 20 and 30, 8; between 30 and 40, 16; between 40 and 50, 32; and for all subsequent trials, 64.

Algorithm 1 summarizes the meta-test procedure. In practice, the gradient-based exploration step is implemented as a update-decode loop. After identifying the current best architecture $\mathcal{G}_*$ in the support set, we compute its embedding $\mathbf{x}_*^{\mathcal{G}}$ and perform a single gradient ascent with $\eta$. We then decode $\tilde{\mathbf{x}}_*^{\mathcal{G}}$ using the graph decoder $\zeta$ to obtain a candidate architecture $\tilde{\mathcal{G}}_*$. If the decoded architecture $\tilde{\mathcal{G}}_*$ is already contained in the current support set $\mathcal{S}$, we repeat the above procedure by recomputing the gradient at the updated $\tilde{\mathbf{x}}_*^{\mathcal{G}}$ and applying another single gradient update followed by decoding. We perform at most 10 such attempts per update-decode loop. If a novel architecture $\tilde{\mathcal{G}}_* \notin \mathcal{S}$ is obtained within 10 attempts, we evaluate it, add it to $\mathcal{S}$, and leave the loop.

## D Comparison of GraB-NAS to Baseline Methods

Table 4 presents the performance of various methods (Random Search, GDAS, SETN, PC-DARTS, DrNAS, MetaD2A, TNAS, and GraB-NAS) on unseen datasets (CIFAR-10, CIFAR-100, MNIST, SVHN, Aircraft, and Pets) during meta-testing.

## E Computational Cost

Table 5 reports a detailed wall-clock time breakdown of the major components of GraB-NAS. Dataset encoding, graph encoding, and graph decoding times are reported as averages per dataset or per architecture, respectively. The GP inference time corresponds to the average cost of predicting the performance of a single candidate architecture on a target dataset using the surrogate.

During the Bayesian Optimization, GP inference must be performed for all candidate architectures. When searching over the NAS-Bench-201 search space, the computational overhead of BO is approximately $0.36\text{ms} \times 15625 = 5.6\text{s}$ per BO step, which is inherent to BO-based methods and is therefore unavoidable. Motivated by this limitation, we introduce the gradient-assisted latent optimization as a lightweight alternative to the BO evaluation.

Table 4: Performance on unseen datasets during meta-test.

| Method | CIFAR-10 Acc (%) | CIFAR-100 Acc (%) | MNIST Acc (%) |
|---|---|---|---|
| Random Search | $90.33 \pm 2.31$ | $64.32 \pm 3.60$ | $99.65 \pm 0.00$ |
| GDAS | $93.61 \pm 0.09$ | $70.70 \pm 0.30$ | $99.64 \pm 0.04$ |
| SETN | $87.64 \pm 0.00$ | $59.09 \pm 0.24$ | $99.69 \pm 0.04$ |
| PC-DARTS | $93.66 \pm 0.17$ | $66.64 \pm 2.34$ | $99.66 \pm 0.04$ |
| DrNAS | $94.36 \pm 0.00$ | $\mathbf{73.51} \pm 0.00$ | $99.59 \pm 0.02$ |
| MetaD2A | $\mathbf{94.37} \pm 0.00$ | $73.44 \pm 0.05$ | $99.67 \pm 0.03$ |
| TNAS | $94.16 \pm 0.11$ | $73.02 \pm 0.12$ | $99.75 \pm 0.00$ |
| GraB-NAS (ours) | $\mathbf{94.37} \pm 0.00$ | $\mathbf{73.51} \pm 0.00$ | $\mathbf{99.78} \pm 0.00$ |

| Method | SVHN Acc (%) | Aircraft Acc (%) | Pets Acc (%) |
|---|---|---|---|
| Random Search | $95.89 \pm 0.35$ | $50.38 \pm 3.19$ | $31.67 \pm 8.81$ |
| GDAS | $95.57 \pm 0.57$ | $53.52 \pm 0.48$ | $24.02 \pm 2.75$ |
| SETN | $96.02 \pm 0.12$ | $44.84 \pm 3.96$ | $25.17 \pm 1.68$ |
| PC-DARTS | $95.40 \pm 0.67$ | $26.33 \pm 3.40$ | $25.31 \pm 1.38$ |
| DrNAS | $96.30 \pm 0.05$ | $46.08 \pm 7.00$ | $26.73 \pm 2.61$ |
| MetaD2A | $96.40 \pm 0.09$ | $55.97 \pm 1.88$ | $39.38 \pm 2.88$ |
| TNAS | $96.54 \pm 0.05$ | $56.14 \pm 0.83$ | $39.12 \pm 0.65$ |
| GraB-NAS (ours) | $\mathbf{96.64} \pm 0.03$ | $\mathbf{58.87} \pm 0.91$ | $\mathbf{43.15} \pm 0.14$ |

Table 5: Wall-clock time breakdown of GraB-NAS components.

| | Time (ms) |
|---|---|
| Dataset Encoding | 2.59 |
| Graph Encoding | 7.08 |
| Graph Decoding | 35.60 |
| GP Inference | 0.36 |
| Gradient Ascent | 14.30 |

As shown in the table, a gradient-based exploration step, including graph encoding, gradient ascent, and graph decoding, requires 57ms in total, which significantly reduces computational cost while preserving strong search effectiveness, as demonstrated in Figure 2.

## F   Evaluation on TransNAS-Bench-101

In this section, we provide additional experimental results on TransNAS-Bench-101 Duan et al. (2021) to evaluate the ability of GraB-NAS. We conduct experiments on the micro search space of TransNAS-Bench-101 using the benchmark data. The full GraB-NAS and the BO-only variant are evaluated on the scene classification task on CIFAR-10, and architectures' performance is measured using test top-1 accuracy. As shown in Figure 6, both variants are capable of identifying competitive architectures. Notably, the full GraB-NAS consistently achieves better any-time performance than the BO-only variant, which demonstrates the effective exploration.

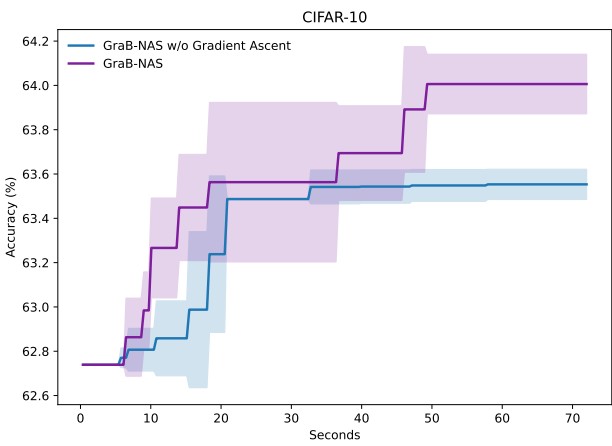

Figure 6: Performance of GraB-NAS evaluated on TransNAS-Bench-101.

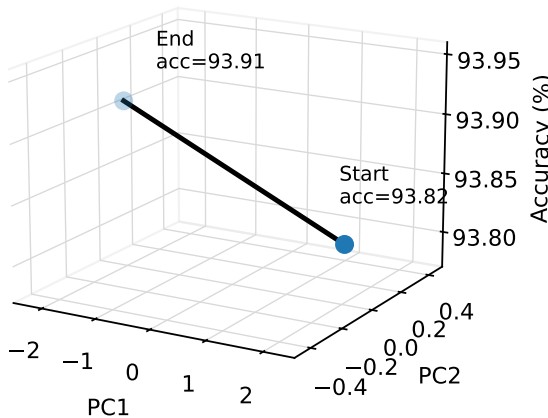

Figure 7: An example of gradient ascent trajectory.

## G   Gradient Ascent Trajectory

In this section, we present the visualization of the gradient ascent trajectory in the latent space. In GraB-NAS, each neural network is represented by a 16-D latent vector learned by the graph encoder. To facilitate the visualization, we project the 16-D latent representations onto a 2-D subspace using PCA Gewers et al. (2021). A representative example of the gradient-based exploration during meta-testing on CIFAR-10 is displayed in Figure 7.

## H   Performance Comparison with TNAS and MetaD2A

In this work, we compare the proposed GraB-NAS with representative Meta-NAS baselines, including TNAS and MetaD2A Shala et al. (2023); Lee et al. (2021). For TNAS, we attempted to reproduce the method using the publicly available code repository. However, the released implementation could not be executed as provided. As a result, we implemented and debugged TNAS based on our best understanding of the algorithm described in the original paper Shala et al. (2023). For MetaD2A, we use the implementation

provided by the authors Lee et al. (2021). To ensure fair comparison, all reported results are averaged over three runs, and we report the mean and standard deviation.

For TNAS, we implement the method by examining both the publicly released codes and the algorithmic description in the original paper Shala et al. (2023). We adapt and debug the released implementation guided by the paper's methodological details. For example, during meta-testing, an additional finetuning step of the deep kernel function within the GP surrogate is present in the provided code. However, in Sections 3.1 and 3.2 of the TNAS paper, it was mentioned that all meta-learned parameters are fixed during meta-testing Shala et al. (2023). The reproduced TNAS results are reported in Figure 2 and Table 4 of our paper, while the original results can be found in Figure 3 and Table 3 of the TNAS paper Shala et al. (2023). We further note that, when accounting for the reported standard deviations, the reproduced TNAS results are lower than the originally reported TNAS numbers on CIFAR-100, Aircraft, and Pets Shala et al. (2023). Such differences may come from the implementation details and experimental setups. Importantly, when considering the standard deviation, the results achieved by GraB-NAS consistently match or exceed the SOTA performance reported in the TNAS paper across the datasets Shala et al. (2023). Beyond the standard meta-test evaluation, we further evaluate both TNAS and GraB-NAS under pruned search space settings, where top-performing architectures are progressively removed. As shown in Table 3, GraB-NAS consistently identifies higher-quality architectures than TNAS, supporting the effectiveness of gradient-based exploration beyond fixed candidate selection.

For MetaD2A, we directly used the publicly available implementation, and reported the results in Figure 2 and Table 4. The original MetaD2A results are reported in Table 1 of the corresponding paper Lee et al. (2021). Our reproduced MetaD2A results are generally consistent with the published numbers on most evaluated datasets Lee et al. (2021). We observe slightly lower performance on the Aircraft dataset, which may be attributed to variance across runs Lee et al. (2021). As for the reported standard deviations, GraB-NAS consistently matches or exceeds the SOTA performances reported by MetaD2A across datasets Lee et al. (2021). As also discussed in TNAS, MetaD2A relies heavily on the meta-training data. and the surrogate model is not updated using feedback from newly evaluated architecture-performance pairs during meta-testing. As a result, performance may stagnate when the target dataset is weakly correlated with the meta-training datasets. In contrast, GraB-NAS continuously incorporates feedback from evaluated architecture-performance pairs into the supporting set, enabling more effective adaptation during meta-testing. This advantage is reflected in our experiments, where GraB-NAS identifies architectures with higher average performance than MetaD2A, as illustrated in Figure 3.

