# OpenReview forum: "Learn to Explore: Meta NAS via Bayesian Optimization Guided Graph Generation"
_TMLR — Rejected by TMLR_

### Review · Reviewer_6WTu · 2025-12-25

**Summary Of Contributions:**

This paper targets the setting of Meta Neural Architecture Search (or Meta NAS), which is to first learn a good prior on some given datasets and pretrained networks and then use the learned priors to enable more efficient searches of neural architectures on other datasets.

The proposed method extends primarily the existing method (TNAS) from Shala et al.. TNAS uses a meta-learned Bayesian surrogate to search architectures for the target task. What this paper proposed is essentially to add an additional component (the graph autoencoder) to TNAS. With the trained graph autoencoder, they propose to perturb the embedding of the found architectures and decode to find additional candidate architectures.

Their evaluation protocol also follows closely (if not exactly) the one from Shala et al. (TNAS) and Lee et al. (MetaD2A): They also use NAS-Bench-201 as search space with meta-training dataset constructed from ImageNet, and they also evaluate performances on the same 6 downstream datasets.

The primary claims are that:
1. The proposed method consistently outperforms existing methods.
2. The introduced gradient-based exploration component (i.e. perturb the embedding of the found architectures and decode to find additional candidate architectures, both through a graph autoencoder) can generate architectures that are not in the original search space.


**reference:**
Gresa Shala, Thomas Elsken, Frank Hutter, and Josif Grabocka. Transfer nas with meta-learned bayesian
surrogates. In The Eleventh International Conference on Learning Representations, 2023.

Lee, Hayeon, Eunyoung Hyung, and Sung Ju Hwang. "Rapid Neural Architecture Search by Learning to Generate Graphs from Datasets." International Conference on Learning Representations. 2021

**Audience:**

Yes

**Audience Explanation:**

Methodology-wise, introducing a graph autoencoder to enable gradient-based update/exploration on model architectures is a plausible idea and I expect that being interesting to at least some TMLR' audience.

**Broader Impact Concerns:**

No notable concern.

**Claims And Evidence:**

No

**Claims Explanation:**

In the experiments presented in the submission, the proposed method GraB-NAS was ranked on top for all 6 downstream datasets, which would have been impressive. However, when looking into the numbers reported in Table 4, I noticed for some baselines (TNAS and MetaD2A specifically) the performance reported are notably lower than the ones reported by Shala et al. and Lee et al.

| target dataset | proposed method | TNAS (reported in this submission) | TNAS (reported by Shala et al.) | MetaD2A (reported in submission) | MetaD2A (reported by Shala et al.) | MetaD2A (reported by Lee et al.) |
|----------------|-----------------|------------------------------------|---------------------------------|----------------------------------|------------------------------------|----------------------------------|
| CIFAR-100      | **73.51**       | 73.02                              | **73.51**                       | 73.44                            | **73.51**                          | **73.51**                        |
| Aircraft       | 58.87           | 56.14                              | **59.51**                       | 55.97                            | 57.71                              | 58.43                            |
| Pets           | 43.15           | 39.12                              | **43.24**                       | 39.38                            | 39.04                              | 41.50                            |


While there may be plausible explanations for such inconsistencies (which one would expect the authors to address), the current evidence is insufficient. In particular, the reported improvements over TNAS in the remaining cases are very marginal (e.g., less than a 0.1% increase in final accuracy), and the TNAS results reported by Shala et al. actually outperform the proposed method in multiple settings. Taken together, these results are insufficient and unconvincing to support the claim that the proposed method consistently outperforms existing approaches.


**reference:**
Gresa Shala, Thomas Elsken, Frank Hutter, and Josif Grabocka. Transfer nas with meta-learned bayesian
surrogates. In The Eleventh International Conference on Learning Representations, 2023.

Lee, Hayeon, Eunyoung Hyung, and Sung Ju Hwang. "Rapid Neural Architecture Search by Learning to Generate Graphs from Datasets." International Conference on Learning Representations. 2021

**Requested Changes:**

My primary concern is about the notably low numbers reported for certain baselines, which is confusing and seriously limits the strength of supports provided to claim the consistent performance gain from the proposed method (in comparison to baselines).

This is critical, and for me to be able to comfortably give a recommendation for acceptance, I would like to see:
1. explanation from authors regarding potential cause of the variation in the reported performances for baselines
2. justification/interpretation regarding why they believe (if they still do, of course) the gain from the proposed method is still significant when factoring in such variations
3. updating the paper accordingly (include responses to the questions above and also likely tune down some claims).

---

> ### Author Response · Authors · 2026-01-16
>
> Dear Reviewer 6WTu,
>
> We thank you for your careful examination of our experimental results and constructive review. Below, we respond to your comments.
>
> **Performance differences**
>
> We thank the reviewer for carefully reviewing the reported baseline results and for pointing out the discrepancies with previously published numbers. We agree that this issue is important and appreciate the opportunity to clarify potential causes.
>
> Regarding TNAS, we attempted to reproduce the reported results using the publicly available implementation. However, the released code could not be executed as provided. We contacted the authors of TNAS to seek clarification, but unfortunately did not receive a response yet. As a result, we implemented the method based on our understanding of the algorithm described in the paper [1], using the publicly available code as a reference. While we carefully followed the reported methodology, minor implementation differences may remain, which could contribute to the observed performance discrepancies.
>
> To further demonstrate the effectiveness and advantages of the proposed GraB-NAS algorithm, we conducted additional experiments under a pruned search space setting, where the top-performing architectures were removed. As shown in the table below (see also Table 3 of the manuscript), **GraB-NAS consistently identifies architectures that surpass the best remaining architectures in the reduced search space. TNAS operates strictly within the predefined search space by design, and its performance is therefore bounded by the best available candidate after pruning.** These results provide complementary evidence that the proposed hybrid search strategy enables GraB-NAS to effectively explore beyond a fixed set of candidate architectures.
>
>
> | Dataset    | Pruning           | Remaining Best | TNAS Acc (Δ ↑)           | GraB-NAS Acc (Δ ↑)        |
> |------------|-------------------|----------------|---------------------------|---------------------------|
> | CIFAR-10   | Top-10 removed    | 94.22          | 94.09 ± 0.09 (−0.13)      | **94.37 ± 0.00 (+0.15)**  |
> |            | Top-20 removed    | 94.12          | 94.02 ± 0.01 (−0.10)      | **94.28 ± 0.13 (+0.16)**  |
> |            | Top-50 removed    | 93.95          | 93.89 ± 0.00 (−0.06)      | **94.22 ± 0.20 (+0.27)**  |
> | CIFAR-100  | Top-10 removed    | 72.88          | 72.82 ± 0.07 (−0.06)      | **73.43 ± 0.12 (+0.55)**  |
> |            | Top-20 removed    | 72.65          | 72.39 ± 0.15 (−0.26)      | **73.20 ± 0.45 (+0.55)**  |
> |            | Top-50 removed    | 72.03          | 71.83 ± 0.24 (−0.20)      | **73.11 ± 0.57 (+1.08)**  |
>
>
> Regarding MetaD2A, the original paper does not specify the random seed [2]. To ensure a fair comparison, we evaluated all methods using the same random seeds and reported the mean and standard deviation over multiple runs. When accounting for the standard deviation, the performance of MetaD2A in our experiments is at a relatively similar level to the results reported in prior work.
>
> [1] Shala G, Elsken T, Hutter F, Grabocka J. Transfer NAS with meta-learned bayesian surrogates. InThe Eleventh International Conference on Learning Representations 2023.
>
> [2] Lee H, Hyung E, Hwang SJ. Rapid neural architecture search by learning to generate graphs from datasets. arXiv preprint arXiv:2107.00860. 2021 Jul 2.

---

> > ### Comment · Reviewer_6WTu · 2026-01-17
> >
> > Hi authors,
> > It is generally understandable that a reimplementation of a baseline may yield performance that differs somewhat from what was previously reported.
> >
> > However, when the reimplemented baseline is notably worse than the original results, this raises a concern that there may be important differences in the setup, implementation, or evaluation protocol that are worth identifying—especially given that your proposed method is built largely on top of this baseline. Since TNAS has publicly released code (https://github.com/TNAS-DCS/TNAS-DCS/tree/main), it would be valuable to the community if you could investigate and clarify the source of the discrepancy.
> >
> > If you are unable to pinpoint the cause, I would at minimum expect the paper to report and discuss the proposed methods by comparing with both the performance of your reimplemented baseline and the numbers reported in the original TNAS & MetaD2A work, with a clear and transparent explanation of the situation. This would help readers properly contextualize the gains of your method.

---

> > > ### Author Response · Authors · 2026-01-23
> > >
> > > Dear Reviewer 6WTu,
> > >
> > > Thank you for the valuable suggestion. We used the code provided at the linked repository to reproduce the TNAS results. However, the released implementation could not reproduce the reported results. We therefore adapted and debugged the implementation based on the released code and our best understanding of the TNAS algorithm described in the paper [1]. Differences between the reproduced and reported results may arise from implementation details and experimental setups. Additional details and analysis are provided in Appendix H of the updated manuscript.
> > >
> > > [1] Shala G, Elsken T, Hutter F, Grabocka J. Transfer NAS with meta-learned bayesian surrogates. InThe Eleventh International Conference on Learning Representations 2023.

---

### Review · Reviewer_N8Qf · 2026-01-01

**Summary Of Contributions:**

This paper proposes GraB-NAS, a Meta-NAS framework that aims to efficiently search for high-performing neural architectures on unseen datasets. It represents architectures as graphs and uses dataset-aware embeddings together with a Gaussian Process surrogate. A hybrid search strategy combines Bayesian Optimization for global exploration and gradient-based optimization in the latent space, enabling the generation of architectures beyond the predefined search space and improving cross-task generalization .

A key strength of the paper is the hybrid search design, which balances global exploration and local refinement and overcomes the limitations of fixed search spaces in prior Meta-NAS methods. The task-aware surrogate model and graph-based architecture generation further enhance generalization and robustness. Extensive experiments and ablation studies provide convincing empirical support for the proposed approach .

One weakness is the reliance on an exact Gaussian Process surrogate, which may limit scalability due to its high computational complexity. In addition, the framework is relatively complex and is evaluated mainly on NAS-Bench-201 and image classification tasks, leaving its applicability to larger or different search spaces unclear .

Overall, GraB-NAS presents a well-designed and effective Meta-NAS framework that achieves strong performance and generalization on unseen tasks, while also pointing to future work on scalability and broader applicability

**Audience:**

Yes

**Audience Explanation:**

The work addresses central research themes in machine learning such as neural architecture search, meta learning, Bayesian optimization, and cross task generalization, all of which align well with the interests of the TMLR community. Researchers working on automated machine learning and efficient model design would find the proposed Meta NAS framework and hybrid search strategy relevant and informative. In addition, the methodological combination of surrogate modeling and gradient based optimization offers insights that may be useful beyond the specific application of neural architecture search.

**Broader Impact Concerns:**

The work does not raise significant ethical concerns that would necessitate an extensive Broader Impact Statement beyond standard discussion. The proposed method focuses on improving the efficiency and generalization of neural architecture search, which is primarily a technical contribution. However, as with other automated machine learning techniques, there is a potential risk that improved automation could lower the barrier to deploying large scale models without sufficient consideration of data bias, environmental cost, or downstream misuse. It would therefore be beneficial for the authors to briefly acknowledge that more efficient architecture search may indirectly contribute to increased model deployment and computational resource usage, and to encourage responsible use of the proposed framework in practice.

**Claims And Evidence:**

Yes

**Claims Explanation:**

The authors provide extensive experimental results on multiple unseen datasets using a standard benchmark (NAS-Bench-201), and consistently compare against strong NAS and Meta-NAS baselines. The improvements in final accuracy, anytime performance, and robustness are clearly demonstrated through quantitative results, rankings, and visual plots. In addition, ablation studies and analyses under pruned search spaces directly support the key claims about the effectiveness of the hybrid search strategy and the ability to generate architectures beyond the predefined search space. While the empirical evidence is strong within the chosen experimental setting, the validation is limited to a specific benchmark and task domain, which slightly narrows the scope of the claims.

**Requested Changes:**

The paper would benefit from several adjustments. First, expanding the experimental evaluation beyond NAS Bench 201 to include larger or more diverse search spaces and tasks would be critical to strengthening the generality of the claims, as it would demonstrate that the proposed method scales and transfers beyond relatively small, cell based benchmarks. Second, a clearer analysis of computational cost and scalability, especially regarding the use of an exact Gaussian Process surrogate during meta testing, would be critical, since efficiency is a central motivation of the approach and a key concern for practical adoption.

In addition, several changes would mainly strengthen the work rather than being strictly required for acceptance. These include providing more implementation details or guidelines to improve reproducibility, offering deeper qualitative analysis or visualizations of the generated architectures to better illustrate the benefits of latent space exploration, and discussing potential failure cases or limitations more explicitly. Clarifying how sensitive the method is to hyperparameters such as the number of BO warm up steps or the gradient ascent learning rate would also help readers better understand and apply the approach.

---

> ### Author Response · Authors · 2026-01-16
>
> Dear Reviewer N8Qf,
>
> We sincerely thank you for your detailed and constructive review. Below, we respond to each of your points in detail.
>
> **Evaluation beyond NAS-Bench-201**
>
> We thank the reviewer for the suggestion. We further extend our evaluation to TransNAS-Bench-101’s micro search space using the benchmark data [1]. Specifically, we train and evaluate the proposed GraB-NAS on the TransNAS-Bench-101 micro search space and conduct architecture search for the scene classification task on CIFAR-10, using test top-1 accuracy as the evaluation metric [1]. We compare two variants (see Figure 6), including the full GraB-NAS and a BO-only GraB-NAS, of the method and report results averaged over three runs. The detailed results are now included in Appendix F in the revised manuscript.
>
> Both variants are able to identify strong architectures, and the full GraB-NAS consistently achieves better anytime performance than the BO-only variant, reaching higher accuracy earlier during the search process. This behavior highlights the advantage of combining global search via BO with local exploration through gradient ascent in the latent space.
>
> Overall, these results further demonstrate the effectiveness of the proposed hybrid search strategy across different search spaces and tasks, reinforcing its applicability beyond the original NAS-Bench-201 setting.
>
> We will include these results and the corresponding discussion to Appendix F in the revised manuscript to further support the generality of the proposed method.
>
> **Computational cost analysis**
> |             | Time (ms) |
> |---------------------|-----------|
> | Dataset Encoding     | 2.59      |
> | Graph Encoding       | 7.08      |
> | Graph Decoding       | 35.60     |
> | GP Inference         | 0.36      |
> | Gradient Ascent      | 14.30     |
>
> To provide a clearer insight into the computational cost, we report a detailed wall-clock time breakdown for each major component of GraB-NAS. Specifically, dataset encoding, graph encoding, and graph decoding times are reported as averages per dataset or per architecture, while GP inference time corresponds to the average prediction cost for evaluating a single candidate architecture on a target dataset.
>
> While GP inference is inherently required to maintain strong prediction performance, its computational complexity becomes substantial when BO requires evaluating the acquisition function over a large search space. For example, in NAS-Bench-201, GP inference cost is approximately **$0.36\text{ms}\times 15,625 \approx 5.6\text{s}$** per BO step, which is unavoidable for BO-based methods. Therefore, we introduce a lightweight gradient-assisted latent optimization to improve the efficiency. A single gradient ascent step, including graph encoding, gradient ascent, and graph decoding, requires **$7.08\text{ms} + 14.30\text{ms} + 35.60\text{ms}\approx 57\text{ms}$**, which is orders of magnitude smaller than the exhaustive GP inference. By performing local optimization in the latent architecture space, GraB-NAS improves search efficiency by avoiding repeated BO evaluations while still enabling effective exploration compared to other Meta-NAS methods solely relying on the BO.
>
> Besides, we would like to emphasize that GraB-NAS is designed as a Meta-NAS framework that generalizes to unseen tasks without retraining, as demonstrated in Figure 2. In this setting, although the meta-test stage is not intended to be extremely lightweight, it is performed without any additional training on the target dataset, which substantially reduces the overall computational burden compared to conventional NAS pipelines. In addition, the proposed GraB-NAS achieves strong anytime performance and attains the best accuracy at any time on some of the evaluated datasets (See Figure 2), which indicates that GraB-NAS is efficient in practice.
>
> We will add these results and the corresponding discussion to Appendix E in the revised manuscript.
>
> **Experimental details**
>
> We thank the reviewer for the valuable suggestions. Further experimental details can be found in Appendix C in the revised manuscript.
>
> [1] Duan Y, Chen X, Xu H, Chen Z, Liang X, Zhang T, Li Z. Transnas-bench-101: Improving transferability and generalizability of cross-task neural architecture search. InProceedings of the IEEE/CVF Conference on Computer Vision and Pattern Recognition 2021 (pp. 5251-5260).

---

### Review · Reviewer_mUaz · 2026-01-01

**Summary Of Contributions:**

The paper proposes GraB-NAS, a Meta-NAS framework designed to generalize across heterogeneous tasks.

Contributions and Strengths
- Combines global architecture search using Bayesian Optimization (BO) with local exploration via gradient ascent in a continuous latent space.
- Unlike many Meta-NAS methods restricted to a fixed search space, GraB-NAS utilizes a graph autoencoder (based on D-VAE) to decode refined latent embeddings into potentially novel architectures.
- Employs a Set Transformer-based dataset encoder to create fused representations of dataset-architecture pairs, which are then processed by a deep kernel Gaussian Process (GP) surrogate to predict performance.
- Demonstrates state-of-the-art (SOTA) performance across six unseen datasets, showing improved generalization and efficiency compared to baselines like MetaD2A and TNAS.

Weaknesses
- GraB-NAS is a combination of D-VAE for graph encoding/decoding, Set Transformers for dataset encoding, and Deep Kernel Learning for the surrogate. The primary novelty is gradient ascent in latent space optimization.
- GraB-NAS did not resolve the core scalability issue. E.g., it still relies on exact GP to maintain competitive performance, which restricts the extensiveness.

**Audience:**

Yes

**Audience Explanation:**

The NAS community is actively looking for ways to break out of fixed search spaces and improve cross-task generalization. The integration of Bayesian Optimization with latent gradient ascent is a logical step forward.

**Broader Impact Concerns:**

There are no significant ethical or broader impact concerns.

**Claims And Evidence:**

Yes

**Claims Explanation:**

I found some of the claims need more clear evidence. For example, the authors claim the ability to generate architectures "beyond the predefined search space". However, the experiments are conducted on NAS-Bench-201, which has a very specific, rigid macro-structure. If the decoder produces a graph that doesn't fit this template, it cannot be evaluated using the NAS-Bench-201 API. If it does fit the template, it is likely already in the search space. The paper fails to provide a single example or statistical analysis of a "novel" architecture discovered that was not in the original space.

**Requested Changes:**

1. Add BO-only Baseline: Include a version of GraB-NAS that uses only Bayesian Optimization to select from the search space, without the gradient-assisted decoder step, to quantify the exploration benefit.
1. Provide a table of wall-clock times for each stage: dataset encoding, GP inference, gradient ascent, and graph decoding.
1. It would be intuitive to provide a visualization of the latent architecture space to show how the gradient ascent trajectory moves from a standard architecture toward a novel high-performing one.

---

> ### Author Response · Authors · 2026-01-16
>
> Dear Reviewer mUaz,
>
> We thank you for the insightful and constructive comments. We appreciate the time and effort devoted to evaluating our work, as well as the constructive suggestions for improving both the clarity and the technical strength of the paper. Below, we provide responses to each comment.
>
> **Additional BO-only Baseline**
>
> We appreciate the reviewer’s suggestion and have added a BO-only ablation variant of GraB-NAS that removes the gradient ascent step and relies solely on Bayesian Optimization (BO) to select architectures. Below are the updated results.
>
> | Method                  | CIFAR-10          | CIFAR-100         |
> |-------------------------|-------------------|-------------------|
> | MetaD2A                 | **94.37 ± 0.00**  | 73.44 ± 0.05      |
> | TNAS                    | 94.16 ± 0.11      | 73.02 ± 0.12      |
> | GraB-NAS w/o Gradient Ascent         | **94.37 ± 0.00**      | 73.43 ± 0.12      |
> | GraB-NAS w/ KNN         | 94.28 ± 0.13      | 73.43 ± 0.12      |
> | GraB-NAS w/ decoder     | **94.37 ± 0.00**  | **73.51 ± 0.00**  |
>
> As shown in the updated results, the GraB-NAS w/o Gradient Ascent performs worse in final accuracy than the full GraB-NAS on both CIFAR-10 and CIFAR-100. This demonstrates that the gradient-assisted latent optimization provides an exploration benefit beyond BO alone. We further note that the differences between TNAS and GraB-NAS w/o Gradient Ascent stem from the design of the algorithm. Specifically, GraB-NAS includes several additional fully connected layers followed by a Softmax function in the MLP-based fusion module that operates on the concatenation of dataset and architecture embeddings.
>
> We will add these results and the corresponding discussion to Section 4.4 in the revised manuscript.
>
> **Wall-clock times for dataset encoding, graph encoding/decoding, gradient ascent, and GP inference**
>
> We thank the reviewer for this valuable suggestion. We provide below a breakdown of the wall-clock time for each major component of GraB-NAS. The reported dataset encoding, graph encoding, and graph decoding times are averaged over a single dataset or architecture. The GP inference time is measured as the average prediction time for evaluating one candidate architecture on a target dataset using the surrogate model. The reported results indicate that the computational overhead introduced by each component of GraB-NAS is small.
>
> |             | Time (ms) |
> |---------------------|-----------|
> | Dataset Encoding     | 2.59      |
> | Graph Encoding       | 7.08      |
> | Graph Decoding       | 35.60     |
> | GP Inference         | 0.36      |
> | Gradient Ascent      | 14.30     |
>
> Notably, the computational cost of selecting the next architecture via BO can be substantial. In NAS-Bench-201, this requires GP inference over all 15,625 architectures, resulting in a total cost of approximately **$0.36\text{ms} \times 15,625 \approx 5.6\text{s}$** per BO step. In contrast, the cost induced by gradient-based exploration (including graph encoding, gradient ascent, and graph decoding) of **$7.08\text{ms} + 14.30\text{ms} + 35.60\text{ms} \approx 57\text{ms}$** is significantly smaller. This comparison highlights that the gradient-based exploration is a lightweight alternative to exhaustive BO-based search, while still enabling effective exploration of the architecture space and facilitating the discovery of higher-performing architectures, as shown in Figure 2.
>
> We will add these results and the corresponding discussion to Appendix E in the revised manuscript.
>
> **Visualization of the gradient ascent trajectory**
>
> We thank the reviewer for suggesting a visualization of the gradient ascent trajectory in the latent architecture space. In GraB-NAS, each architecture is represented by a 16-D latent vector learned by the graph encoder. To enable visualization, we project these latent representations onto a 2-D subspace using Principal Component Analysis (PCA), which preserves the dominant variance of the trajectory while providing an interpretable low-dimensional view. We provide an example (see Figure 7) of the gradient-based exploration during the meta-test on CIFAR-10. Note that the ‘Accuracy’ in the figure corresponds to the evaluated accuracy.
>
> We will add the example to Appendix G in the revised manuscript.

---

### Decision · Action_Editor_oRuu · 2026-03-08

**Recommendation:** Reject

**Additional Comments:**

I recommend Reject (with resubmission allowed after major revisions).
The submission is technically sound in its framework design but falls short of TMLR’s “sound” criterion due to the unresolved experimental issues outlined above. To make the work acceptable upon resubmission, the authors must:

Substantially expand and systematize the material currently in Appendix H (or promote the most critical parts to the main paper).

Strengthen the empirical grounding for “novel architecture” claims by providing at least one set of experiments that is not strictly constrained by NAS-Bench-201’s API.

Address the scalability limitations of exact GPs more explicitly and, if possible, include early results or analysis on approximate GPs for larger problems.

These revisions are concrete, feasible, and necessary to turn promising ideas into convincingly supported contributions. I do not recommend any TMLR certification at the present stage, as the current evidence does not yet justify such recognition.

**Audience:**

Yes

**Audience Explanation:**

The topic of meta neural architecture search that combines Bayesian optimization with gradient-based latent-space exploration, while using dataset encoders and graph autoencoders to move beyond fixed search spaces, directly aligns with active interests in automated ML, NAS efficiency, and few-shot generalization. Researchers working on Bayesian optimization, deep kernel GPs, or practical NAS for unseen tasks (CIFAR, SVHN, fine-grained benchmarks) would find value in the high-level ideas, even if the current empirical validation requires strengthening.

**Claims And Evidence:**

No

**Claims Explanation:**

The core claims of GraB-NAS—namely consistent outperformance over strong baselines (MetaD2A, TNAS, etc.) in search effectiveness, efficiency, generalization, and the ability to generate truly novel task-aware architectures beyond fixed search spaces—are not fully supported by accurate, convincing, and clear evidence. While the main experiments appear promising and two reviewers accepted the claims at face value, my close examination of Appendix H (the section added to address low reimplemented baseline results) shows that the supplementary material is incomplete and lacks systematic comparisons. Without these, the reported performance gains cannot be confidently attributed to the proposed method rather than implementation artifacts or benchmark constraints. This renders the evidence unconvincing on key claims.

**Resubmission Of Major Revision:**

The authors may consider submitting a major revision at a later time.